# Risk of gastrointestinal cancer in patients with an elevated level of gamma-glutamyltransferase: A nationwide population-based study

Seung Wook Hong[1]¤, Hyun Jung Lee[1]*, Kyungdo Han[2], Jung Min Moon[1], Seona Park[1], Hosim Soh[1], Eun Ae Kang[1], Jaeyoung Chun[3], Jong Pil Im[1], Joo Sung Kim[1]

1 Department of Internal Medicine and Liver Research Institute, Seoul National University College of Medicine, Seoul, Korea, 2 Department of Medical Statistics, The Catholic University of Korea College of Medicine, Seoul, Korea, 3 Department of Internal Medicine, Gangnam Severance Hospital, Yonsei University College of Medicine, Seoul, Korea

¤ Current address: Department of Gastroenterology, Asan Medical Center, University of Ulsan College of Medicine, Seoul, South Korea
* guswjd80@gmail.com

**Data Availability Statement:** The data underlying the results presented in the study are available from the National Health Insurance Sharing Service. Detailed instructions for the process of

## Abstract

Emerging evidence that an elevated serum gamma-glutamyltransferase (GGT) level is associated with an increased risk of gastrointestinal cancer, but still controversial. The aim of this study to assess the relationship between GGT level and risk of gastrointestinal cancer, and the contribution of the interaction of hyperglycemia with elevated GGT level to the incidence of gastrointestinal cancer by the stratified analysis. A total of 8,120,665 Koreans who received medical checkups in 2009 were included. Subjects were classified according to the quartile of GGT level for women and men. The incidence rates of gastrointestinal cancer for each group were analyzed using Cox proportional hazards models. During follow-up, 129,853 cases of gastrointestinal cancer newly occurred (esophagus, 3,792; stomach, 57,932; and colorectal, 68,789 cases). The highest GGT quartile group showed an increased risk of gastrointestinal cancer (esophagus, hazard ratio = 2.408 [95% confidence interval, 2.184–2.654]; stomach, 1.121 [1.093–1.149]; and colorectal, 1.185 [1.158–1.211]). The risk increased significantly with the rise in GGT quartile level, regardless of the site of cancer. The stratified analysis according to glycemic status showed that the effect of elevated GGT was predominant in the risk of esophageal cancer. The effect of elevated GGT further increased the risk of stomach and colorectal cancers in diabetic patients. An elevated level of GGT was associated with an increased risk of gastrointestinal cancer, regardless of the site of cancer. The effect of the increase in GGT level on the risk of gastrointestinal cancer depended on the type of cancer and glycemic status.

## Introduction

Serum gamma-glutamyltransferase (GGT) is an indicator related to hepatic dysfunction and used as a surrogate marker that reflects excessive alcohol consumption [1, 2]. Several epidemiologic studies have reported that elevated GGT level is associated with the incidence of various

receiving authorization to access the data can be found here: https://nhiss.nhis.or.kr/bd/ab/bdaba032eng.do. The authors of this study did not receive any special privileges in accessing the data.

**Funding:** The author(s) received no specific funding for this work.

**Competing interests:** The authors have declared that no competing interests exist.

disease including metabolic syndrome, diabetes mellitus (DM), and cardiovascular disease [3–9]. Emerging evidence indicates that an elevated GGT level is also linked to increased risk of cancer. Notably, many studies have demonstrated a positive correlation between an elevated GGT level and the risk of gastrointestinal cancer, but results have been inconsistent depending on sex and the site of cancer [10–16]. Hence, further study with a large population is needed to establish any association between elevated GGT level and risk of gastrointestinal cancer.

Diabetes mellitus is a chronic disease of increasing prevalence, generally accompanied by hyperglycemia caused by insulin resistance [17]. An elevated GGT level is related to hyperglycemia, and many studies have reported that patients with DM have an increased risk of gastrointestinal cancer [18–24]. However, few studies have assessed the association of serum GGT level and glycemic status with the incidence of gastrointestinal cancer in the general population, and little is known about whether the effect of GGT on the gastrointestinal cancer risk is influenced by glycemic status.

Therefore, we aimed to assess whether an elevated serum GGT level is associated with the risk of gastrointestinal cancer through a large-scale study using nationwide data. In addition, we evaluated the effect of glycemic status on the association between an elevated GGT level and the risk of gastrointestinal cancer, and the contribution of the interaction between these two factors on the incidence of gastrointestinal cancer.

## Materials and methods

### Data sources

This population-based study was retrospectively conducted using the database from the National Health Insurance Service (NHIS) in South Korea. The NHIS is one of the social security systems provided by the government, and all Koreans are obliged to enroll in. It serves to pay health care providers for medical services on behalf of the general population. All data related to the medical services guaranteed by the NHIS are stored in this database. The NHIS database collects demographic data as well as the information regarding medical services, including medications, hospitalization, and diagnoses identified by the International Classification of Disease, Tenth Revision (ICD-10) codes of the subscribers. The NHIS recommends that subscribers undergo a general medical check-up biennially. This general medical check-up includes body measurements, chest X-ray, and blood chemistry tests.

### Data collection and study population

Demographic information including age, sex, and income was stored in the NHIS database for all participants who underwent general medical checkups. The weight, height, and blood pressure for participants were measured, and laboratory tests including fasting blood glucose (FBG), aspartate aminotransferase, alanine aminotransferase, GGT level, and cholesterol were performed on the same day. Body mass index (BMI) was calculated by dividing the weight (kg) by height squared ($m^2$). A self-reporting questionnaire was used to assess drinking habits, history of smoking, income, and exercise habits for the participants. The history of smoking was classified into three categories: (1) non-smoker; (2) ex-smoker; and (3) current smoker. An ex-smoker was defined as a person with smoking experience of >100 cigarettes, although smoking had currently ceased. A heavy drinker was defined as a person drinking >30 g of alcohol per week.

Adults over the age of 20 years who underwent a general medical check-up provided by the NHIS during January–December 2009 qualified for registration in this cohort. At the time of enrolment, patients who were previously diagnosed with cancer identified by ICD-10, regardless of the type, were excluded. To minimize disruption of pre-existing factors to the incidence

of gastrointestinal cancer, any type of cancer patients who were diagnosed within one year of the time of enrolment were excluded from the cohort. Individuals diagnosed with liver cirrhosis (K703) or hepatitis (K746) were also excluded. To avoid the confounding effect of excessive alcohol consumption on the serum GGT level, individuals with heavy drinking were excluded from the cohort.

## Outcomes and ethical considerations

Since there might be a variability of GGT levels depending on measurement equipment and no threshold was reported to be associated with an increased risk of gastrointestinal cancer, subjects in the cohort were divided into four groups according to the quartile level of GGT. The quartile level of GGT was differently determined, considering differences in the level of GGT by sex (men: first quartile [Q1], <21 U/L; second quartile [Q2], 21–29 U/L; third quartile [Q3], 30–47 U/L; and fourth quartile [Q4], ≥48 U/L; and women: Q1, <13 U/L; Q2, 13–15 U/L; Q3, 16–22 U/L; and Q4, ≥23 IU/L). The primary outcome was to evaluate the incidence rate of gastrointestinal cancer for each group by the quartile level of GGT. The secondary outcome was indirectly to assess the interaction of an elevated GGT level and hyperglycemia contributing to the risk of gastrointestinal cancer, using stratified analysis according to glycemic status. The glycemic status was categorized in accordance with FBG (non-DM, FBG < 100 mg/dL; impaired fasting glucose (IFG), 100 ≤ FBG < 126 mg/dL; and DM, FBG ≥ 126 mg/dL). The follow-up was terminated at the occurrence of gastrointestinal cancer and monitoring of other members of the cohort was continued to until 31 December 2017. The gastrointestinal cancer included esophageal, stomach, and colorectal cancers. Cases of newly diagnosed gastrointestinal cancer during the follow-up period were identified using ICD-10 codes in the NHIS database as follows–esophageal (C15), stomach (C16), and colorectal cancer (C18–20)–and their index date was the first day that cancer was recorded. This study was approved by the Institutional Review Board at Seoul National University Hospital. All data were fully anonymized before we access them, so the requirement for informed consent was waived. (IRB approved data: 2019.06.07, No. of IRB: E-1906-008-1036).

## Statistical analysis

Data were expressed as mean ± standard deviation for continuous variables and as number and proportions for categorical variables. To compare differences among groups, one-sided ANOVA were used for continuous data and the chi-square test for categorical data. The incidence rate of gastrointestinal cancer was calculated by dividing the number of incident cases by the overall follow-up period and presented as 10,000 person-years. The Cox proportional hazard model was used to evaluate the association GGT level and risk of gastrointestinal cancer. In addition, stratified analyses were conducted according to glycemic status. Two adjustment models were applied, model 1 was adjusted for age and sex, and model 2 was adjusted for age, sex, BMI, smoking, drinking, exercise, and income which were known to be risk factors for gastrointestinal cancer. A statistical significance was defined as $P < 0.05$. Statistical analyses were performed using SAS version 9.4 (SAS Institute, Cary, NC, USA).

## Results

### Baseline characteristics

A total of 8,120,665 individuals were enrolled in this cohort. The selection process for the study subjects is illustrated in Fig 1. The eligible population was categorized into four groups by the quartile level of serum GGT. The baseline characteristics of the population in

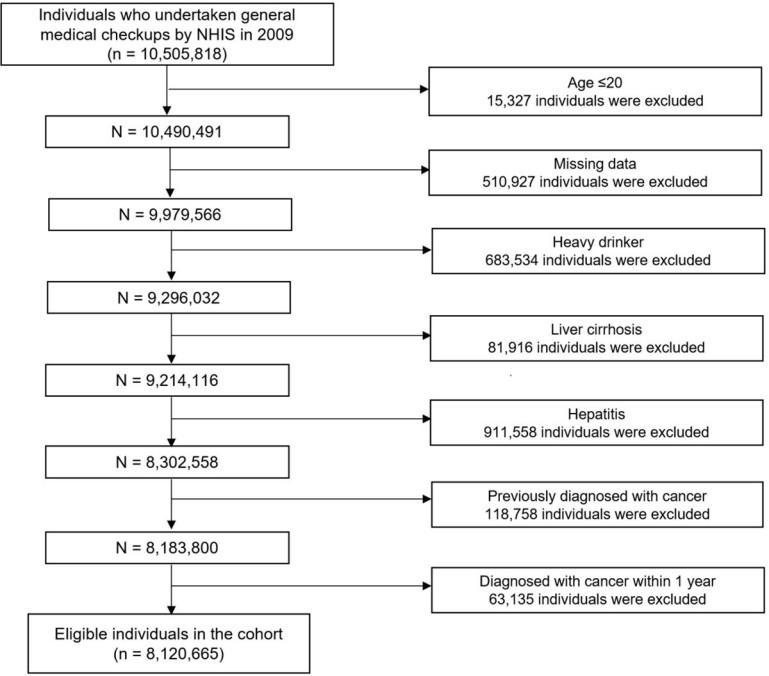

**Fig 1. Flow chart of selection process of study subject.**

accordance with GGT quartiles are presented in Table 1. Compared with the lower GGT quartile groups (Q1–3), more individuals who drank or smoked were included in the higher GGT quartile group (Q4). The proportion of subjects with hypertension, dyslipidemia, and

**Table 1. Baseline characteristics by serum GGT level (n = 8,120,665).**

| | GGT quartile | | | | P-value |
|---|---|---|---|---|---|
| | Q1 | Q2 | Q3 | Q4 | |
| | N = 2,001,551 | N = 2,029,189 | N = 2,099,258 | N = 1,990,667 | |
| Age (years)* | 43.75 ± 14.75 | 45.93 ± 14.36 | 47.87 ± 13.83 | 49.28 ± 12.9 | < .0001 |
| Sex, Men (%) | 1,047,272 (52.32) | 1,026,998 (50.61) | 1,105,896 (52.68) | 1,054,248 (52.96) | < .0001 |
| Waist circumference (cm)* | 76.34 ± 8.1 | 78.36 ± 8.66 | 80.92 ± 8.91 | 83.5 ± 8.88 | < .0001 |
| Body mass index (kg/m²)* | 22.37 ± 2.71 | 23.1 ± 2.96 | 24.02 ± 3.16 | 24.97 ± 3.34 | < .0001 |
| Exercise (%) | 1,036,622 (51.79) | 1,044,128 (51.46) | 1,074,483 (51.18) | 1,000,684 (50.27) | < .0001 |
| Smoking status (%) | | | | | < .0001 |
| Non-smoker | 1,330,105 (66.45) | 1,308,008 (64.46) | 1,272,410 (60.61) | 1,140,435 (57.29) | |
| Ex-smoker | 259,333 (12.96) | 269,649 (13.29) | 294,621 (14.03) | 261,707 (13.15) | |
| Current smoker | 412,113 (20.59) | 451,532 (22.25) | 532,227 (25.35) | 588,525 (29.56) | |
| Low income (%) | 425,064 (21.24) | 429,320 (21.16) | 440,156 (20.97) | 425,771 (21.39) | < .0001 |
| Drinker (%) | 783,461 (39.14) | 876,698 (43.2) | 997,037 (47.49) | 1,060,510 (53.27) | < .0001 |
| Hypertension (%) | 290,349 (14.51) | 403,676 (19.89) | 564,871 (26.91) | 713,722 (35.85) | < .0001 |
| Dyslipidemia (%) | 165,360 (8.26) | 272,186 (13.41) | 420,969 (20.05) | 577,006 (28.99) | < .0001 |
| Metabolic syndrome (%) | 218,539 (10.92) | 371,250 (18.3) | 607,707 (28.95) | 869,756 (43.69) | < .0001 |
| Fasting glucose (mg/dl)* | 92.13 ± 16.49 | 94.07 ± 18.82 | 97.15 ± 22.32 | 102.55 ± 27.88 | < .0001 |

GGT, gamma-glutamyltransferase.

*Value was presented with mean ± SD.

metabolic syndrome gradually increased as the GGT quartile level increased. The level of FBG also increased as the level of GGT quartile increased. The baseline characteristics of included population according to the GGT quartile level after stratification by sex are shown in S1 and S2 Tables.

## Association between serum GGT level and the risk of gastrointestinal cancer

During the 8-year follow-up, 3792 cases of esophageal cancer, 57,932 cases of stomach cancer, and 68,789 cases of colorectal cancer developed. For each type of gastrointestinal cancer, the number of incident cases, the incidence rate, and the adjusted hazard ratio (HR) according to the GGT quartile level are shown in Table 2. After adjusting for age, sex, BMI, smoking, drinking, exercise, and income, the risk of gastrointestinal cancer significantly increased with a gradual rise of the serum GGT level, regardless of the type of cancer. The adjusted HRs for esophageal cancer incidence for Q2, Q3, and Q4 in comparison with Q1 were 1.218 [95% confidence interval (95%CI), 1.098–1.351], 1.445 [1.305–1.559], and 2.408 [2.184–2.654], respectively (*p* for trend <0.001). The adjusted HRs for stomach cancer incident for Q2, Q3, and Q4 in comparison with Q1 were 1.012 [0.987–1.037], 1.042 [1.017–1.068], and 1.121 [1.093–1.149], respectively (*p* for trend <0.001). Similarly, the adjusted HRs for colorectal cancer incidence for Q2, Q3, and Q4 in comparison with Q1 were 1.030 [1.007–1.054], 1.091 [1.067–1.116], and 1.185 [1.158–1.212], respectively (*p* for trend <0.001) (Fig 2).

## Association between hyperglycemia and risk of gastrointestinal cancer

For each type of gastrointestinal cancer, the number of incident cases and incidence rates stratified by glycemic status are presented in Table 3. After multivariable adjustment, compared to the non-DM group, the incidence of each gastrointestinal cancer was higher in both IFG and DM group (esophagus, HR: 1.198 [1.113–1.290] in the IFG group and 1.278 [1.170–1.397] in

**Table 2. Risk of gastrointestinal cancer according to serum GGT quartile level.**

| Cancer site | | Q1 | Q2 | Q3 | Q4 | P for trend |
|---|---|---|---|---|---|---|
| | | | GGT quartile | | | |
| **Esophagus** | Cases, n | 671 | 780 | 945 | 1,396 | |
| | Incidence rate[1] | 0.46 | 0.53 | 0.62 | 0.97 | |
| | Model 1, HR (95% CI)[2] | 1* | 1.194 (1.077–1.324) | 1.403 (1.270–1.549) | 2.415 (2.200–2.651) | <0.001 |
| | Model 2, HR (95% CI)[3] | 1* | 1.218 (1.098–1.351) | 1.445 (1.305–1.559) | 2.408 (2.184–2.654) | <0.001 |
| **Stomach** | Cases, n | 12,406 | 13,530 | 15,470 | 15,886 | |
| | Incidence rate[1] | 8.52 | 9.17 | 10.15 | 11.04 | |
| | Model 1, HR (95% CI)[2] | 1* | 1.028 (1.003–1.053) | 1.073 (1.048–1.099) | 1.172 (1.144–1.200) | <0.001 |
| | Model 2, HR (95% CI)[3] | 1* | 1.012 (0.987–1.037) | 1.042 (1.017–1.068) | 1.121 (1.093–1.149) | <0.001 |
| **Colorectal** | Cases, n | 13,749 | 15,802 | 18,953 | 20,265 | |
| | Incidence rate[1] | 9.45 | 10.72 | 12.45 | 14.09 | |
| | Model 1, HR (95% CI)[2] | 1* | 0.983 (0.918–1.052) | 1.075 (1.010–1.145) | 1.209 (1.139–1.284) | <0.001 |
| | Model 2, HR (95% CI)[3] | 1* | 1.03 (1.007–1.054) | 1.091 (1.067–1.116) | 1.185 (1.158–1.212) | <0.001 |

GGT, gamma-glutamyltransferase.

*Reference value.

[1]Per 10,000 person-year

[2]Adjusted for age, sex

[3]Adjusted for age, sex, BMI, smoking, drinking, exercise, income.

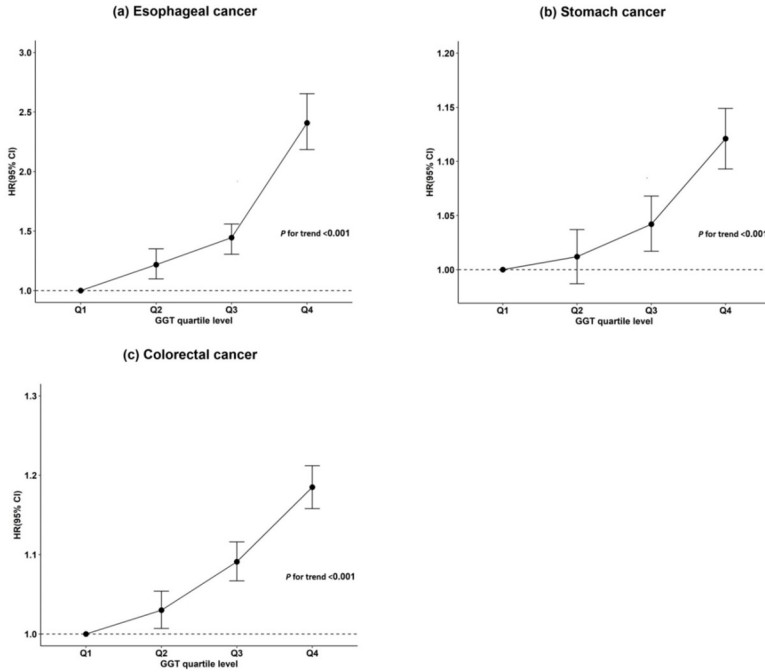

**Fig 2. Association between serum GGT quartile level and risk of gastrointestinal cancer.** (a) esophageal cancer, (b) stomach cancer, and (c) colorectal cancer. Q1-4: GGT quartile level.

the DM group; stomach: 1.050 [1.030–1.071] in the IFG group and 1.181 [1.154–1.210] in the DM group; and colorectal: 1.089 [1.069–1.108] in the IFG group and 1.217 [1.191–1.244] in the DM group).

**Table 3. Risk of gastrointestinal cancer according to glycemic status.**

| Cancer site | | Glycemic status | | | P for trend |
| --- | --- | --- | --- | --- | --- |
| | | Non-DM | IFG | DM | |
| Esophagus | Cases, n | 1,996 | 1,122 | 674 | |
| | Incidence rate[1] | 0.48 | 0.87 | 1.44 | |
| | Model 1, HR (95% CI)[2] | 1* | 1.155 (1.074–1.243) | 1.168 (1.070–1.276) | <0.001 |
| | Model 2, HR (95% CI)[3] | 1* | 1.198 (1.113–1.290) | 1.278 (1.170–1.397) | <0.001 |
| Stomach | Cases, n | 32,862 | 18,214 | 10,951 | |
| | Incidence rate[1] | 7.93 | 11.74 | 20.03 | |
| | Model 1, HR (95% CI)[2] | 1* | 1.056 (1.036–1.077) | 1.191 (1.164–1.220) | <0.001 |
| | Model 2, HR (95% CI)[3] | 1* | 1.050 (1.030–1.071) | 1.181 (1.154–1.210) | <0.001 |
| Colorectal | Cases, n | 39,604 | 18,214 | 10,951 | |
| | Incidence rate[1] | 9.57 | 14.14 | 23.60 | |
| | Model 1, HR (95% CI)[2] | 1* | 1.109 (1.089–1.128) | 1.246 (1.220–1.274) | <0.001 |
| | Model 2, HR (95% CI)[3] | 1* | 1.089 (1.069–1.108) | 1.217 (1.191–1.244) | <0.001 |

DM, Diabetes mellitus; IFG: Impaired fasting glucose.

*Reference value.

[1]Per 10,000 person-year

[2]Adjusted for age, sex

[3]Adjusted for age, sex, BMI, smoking, drinking, exercise, income.

## Effect of serum GGT level on the risk of gastrointestinal cancer stratified by glycemic status

Stratified analyses by glycemic status were conducted to evaluate the interaction of an elevated GGT level and hyperglycemia contributing to the risk of gastrointestinal cancer. The subjects were stratified into non-DM, IFG, and DM groups by glycemic status, and each group was divided into Q4 groups and other groups (Q1–3) in accordance with the serum GGT quartile level. After setting Q1–3 groups in non-DM as a reference group, the adjusted HR of the incidence in gastrointestinal cancer was estimated for each group (Table 4).

An elevated GGT level had a significant effect on the incidence of esophageal cancer. The risk of esophageal cancer was higher in the Q4 than Q1–3 groups regardless of glycemic status. Additionally, the Q4 groups of non-DM and IFG showed a higher risk of esophageal cancer than the Q1–3 group of DM (non-DM/Q4, HR: 1.98 [95%CI, 1.799–2.180]; IFG/Q4, 2.089 [1.873–2.331]; and DM/Q1-3, 1.168 [1.037–1.316]) (Fig 3A). An elevated serum GGT level and hyperglycemia showed an interactive effect on the incidence of stomach and colorectal cancer. For stomach cancer, the Q4 group of DM showed the highest risk (HR: 1.283 [95%CI: 1.237–1.331]), and the HR did not differ between the Q4 group of IFG and the Q1–3 group of DM (IFG/Q4, HR: 1.129 [95%CI: 1.094–1.165]; and DM/Q1-3, 1.164 [1.130–1.199]) (Fig 3B). The risk of colorectal cancer was highest in the Q4 group of DM (HR: 1.342 [95%CI: 1.299–1.386]). In contrast, the HR in the Q4 group of IFG was higher than that in the Q1–3 group of DM, although not significant (IFG/Q4, HR: 1.214 [95%CI: 1.180–1.248]; and DM/Q1-3, 1.193 [1.161–1.227]) (Fig 3C).

## Discussion

In this nationwide population-based study, we demonstrated that an elevated serum GGT level was significantly associated with the risk of gastrointestinal cancer. In regard to site-specific cancer incidence, the risks of esophageal, stomach, and colorectal cancers all increased as the GGT level increased. Moreover, the risk of gastrointestinal cancer was also related to hyperglycemia including IFG. The stratified analysis of the association between an elevated GGT level and the risk of gastrointestinal cancer by glycemic status showed that an elevated GGT level had a dominant effect on incidence of esophageal cancer. Notably, an elevated GGT level and hyperglycemia showed an interactive effect on the incidence of stomach and colorectal cancers. To the best of our knowledge, this is the first large-scale study to confirm the

**Table 4. Association risk of gastrointestinal cancer and serum GGT level stratified by glycemic status.**

| Glycemic status | GGT level | Cancer site | | | | | |
| --- | --- | --- | --- | --- | --- | --- | --- |
| | | Esophagus | | Stomach | | Colorectal | |
| | | Incidence rate[1] | HR (95% CI)[2] | Incidence rate[1] | HR (95% CI)[2] | Incidence rate[1] | HR (95% CI)[2] |
| **Non-DM** | Q1-Q3 | 0.41 | 1* | 7.68 | 1* | 9.11 | 1* |
| | Q4 | 0.76 | 1.98 (1.799–2.180) | 9.02 | 1.077 (1.049–1.105) | 11.44 | 1.093 (1.068–1.119) |
| **IFG** | Q1-Q3 | 0.77 | 1.193 (1.088–1.309) | 11.71 | 1.043 (1.019–1.067) | 13.61 | 1.067 (1.044–1.090) |
| | Q4 | 1.09 | 2.089 (1.873–2.331) | 11.90 | 1.129 (1.094–1.165) | 15.41 | 1.214 (1.180–1.248) |
| **DM** | Q1-Q3 | 1.32 | 1.168 (1.037–1.316) | 21.51 | 1.164 (1.130–1.199) | 24.17 | 1.193 (1.161–1.227) |
| | Q4 | 1.60 | 2.341 (2.063–2.656) | 18.01 | 1.283 (1.237–1.331) | 22.85 | 1.342 (1.299–1.386) |

DM: Diabetes Mellitus, IFG: Impaired fasting glucose.

*Reference value.

[1] Per 10,000 person-year

[2] Adjusted for age, sex, BMI, smoking, drinking, exercise, income.

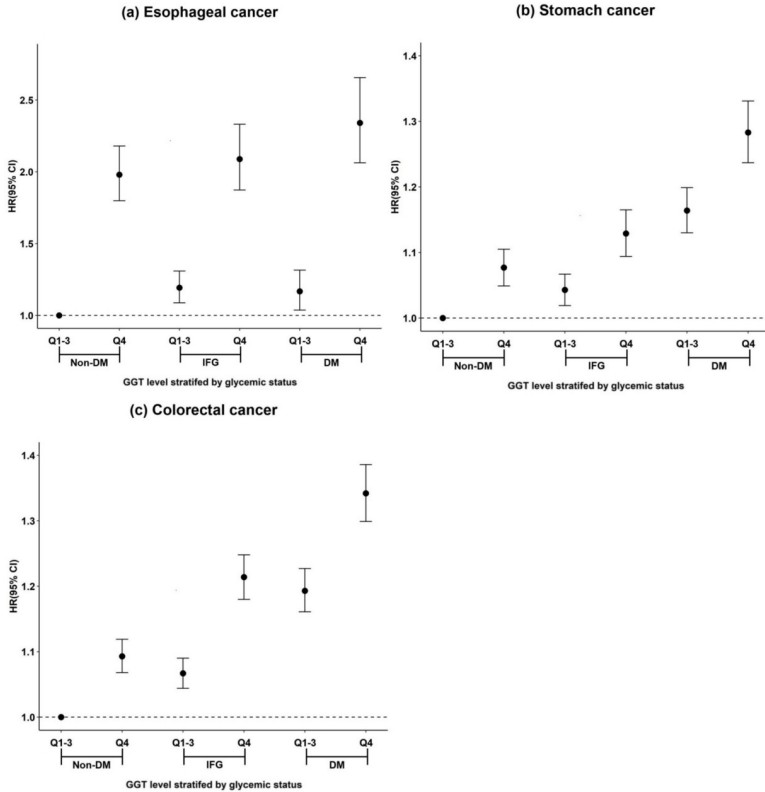

**Fig 3. Effect of an elevated GGT level on the risk of gastrointestinal cancer stratified by glycemic status.** (a) esophageal cancer, (b) stomach cancer, and (c) colorectal cancer. DM: diabetes mellitus IFG: impaired fasting glucose, Q1-4: GGT quartile level.

association of elevated GGT level and hyperglycemia with the risk of gastrointestinal cancer and to reveal the interaction of the two factors.

Our findings are consistent with previous several epidemiologic studies reporting a significant correlation between an elevated GGT level and incidence of gastrointestinal cancer. Cohort studies from Austria and Sweden revealed that subjects with high serum GGT levels had an increased risk of gastrointestinal cancer [10, 12, 13], and a meta-analysis found a positive association between serum GGT levels and the risk of gastrointestinal cancer [15]. Another study in Korea showed that an elevated GGT level was associated with an increased risk of esophageal, stomach, and colorectal cancers, but this was significant only in men [14]. Although the underlying mechanism for the link between an elevated GGT level and the risk of gastrointestinal cancer is not completely understood, an elevated serum GGT level may increase the risk of cancer via the oxidative stress pathway related to glutathione metabolism [25]. Several preclinical studies identified that increased GGT activity can be a response to oxidative stress and, in turn, the persistent production of reactive oxygen species by increased GGT activity may cause genetic instability and tumor progression [25–29]. These findings are the most plausible mechanisms related to the association between an elevated GGT level and increased risk of gastrointestinal cancer.

Although there was a difference in degree for the specific site of cancer, an elevated GGT level and hyperglycemia were both related to the increased risk of gastrointestinal cancer and the risk was highest in the high GGT group of DM patients. Moreover, it is well known that an elevated GGT level is associated with hyperglycemia because several studies found that an

elevated GGT level may be involved in the development of prediabetes or diabetes [30–32]. There is corroborative evidence to support the correlations between pairs among an elevated GGT level, hyperglycemia, and carcinogenesis. Insulin resistance is a pivotal mechanism to develop prediabetes and type 2 DM, and it causes hyperinsulinemia [17]. Although it is still uncertain, hyperinsulinemia may promote cell proliferation via various pathways and ultimately contribute to the incidence of cancer [33–35]. In addition, increased oxidative stress may induce insulin resistance and pancreatic β-cell dysfunction [36, 37]. Chronic inflammation is closely related to both insulin resistance and increased oxidative stress and has been recognized as a critical factor for carcinogenesis [38, 39]. It is presumed that a series of these factors interact with one another, affecting the development of cancer. Further studies are needed to investigate the cause of differences in the interaction between an elevated GGT level and hyperglycemia according to the type of cancer, and the fundamental mechanism of carcinogenesis.

It is remarkable that the impact of an elevated GGT level and hyperglycemia on the incidence of gastrointestinal cancer differed with the specific site of cancer. The effect of high GGT level was overwhelming in developing esophageal cancer. However, the interactive effect of an elevated GGT level and hyperglycemia, which contributed to the risk of stomach and colorectal cancer, was indirectly revealed. Considering the higher tendency for colorectal cancer risk in patients with high GGT level of the IFG group compared with a low GGT level of the DM group, the serum GGT level has potential as a discriminating marker in predicting colorectal cancer, especially in prediabetic patients. Therefore, development of a model is required to precisely estimate an individual's risk of developing gastrointestinal cancer based on serum GGT level and glycemic status.

This was a large-scale cohort study with nearly 8.1 million participants. However, this study had several limitations. First, it was limited to assessing the effect of lowering the GGT level or FBG on the incidence of gastrointestinal cancer, because only the baseline levels were collected. Further longitudinal studies are needed to evaluate the change in the incidence of gastrointestinal cancer as the level of GGT changes. In addition, the definition of DM in this study did not include all the diagnostic criteria for DM. Although this definition has been widely used in previous nationwide studies [40, 41], additional studies with a more reliable definition for DM are required. To dispel concern due to the confounding effect of hepatic dysfunction on the outcomes, we excluded patients with liver cirrhosis or hepatitis. However, the results for liver function data including of liver enzymes were not adjusted. Finally, the results on the interaction between an elevated GGT level and glycemic status should be interpreted cautiously because they were estimated with the indirect statistical method of stratified analysis.

In conclusion, this large-scale study showed that an elevated GGT level was associated with an increased risk of gastrointestinal cancer. Additionally, the effect of an elevated GGT level on the cancer incidence varied with the type of cancer and glycemic status. Further studies are needed to assess the effect of sequential change in GGT level and glycemic status on the incidence of gastrointestinal cancer. Despite some limitations, this study provides meaningful evidence for clinicians to develop a strategy for screening of gastrointestinal cancer according to GGT level and glycemic status.

## Supporting information

**S1 Table. Baseline characteristic by serum GGT level in men ($n$ = 4,234,414).** (DOCX)

**S2 Table. Baseline characteristic by serum GGT level in women ($n$ = 3,886,251).**
(DOCX)

## Author Contributions

**Conceptualization:** Seung Wook Hong, Hyun Jung Lee, Jung Min Moon, Seona Park, Hosim Soh, Jaeyoung Chun, Jong Pil Im, Joo Sung Kim.

**Data curation:** Kyungdo Han.

**Formal analysis:** Kyungdo Han.

**Investigation:** Seung Wook Hong.

**Methodology:** Seung Wook Hong, Kyungdo Han.

**Project administration:** Hyun Jung Lee.

**Resources:** Seung Wook Hong.

**Supervision:** Hyun Jung Lee, Jung Min Moon, Seona Park, Hosim Soh, Eun Ae Kang, Jaeyoung Chun, Jong Pil Im, Joo Sung Kim.

**Visualization:** Seung Wook Hong.

**Writing – original draft:** Seung Wook Hong.

**Writing – review & editing:** Hyun Jung Lee, Jung Min Moon, Seona Park, Hosim Soh, Eun Ae Kang, Jaeyoung Chun, Jong Pil Im, Joo Sung Kim.

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
