## [Decision Letter · Decision Letter 0]

29 Sep 2020

PONE-D-20-12359

Risk of Gastrointestinal Cancer in Patients with an Elevated Level of Gamma-glutamyltransferase: A Nationwide Population-Based Study.

PLOS ONE

Dear Dr. Hyun Jung Lee,

Thank you for submitting your manuscript to PLOS ONE. After careful consideration, we feel that it has merit but does not fully meet PLOS ONE’s publication criteria as it currently stands. Therefore, we invite you to submit a revised version of the manuscript that addresses the points raised during the review process.

The reviewers have raised a number of points which we believe major modifications are necessary to improve the manuscript, taking into account the reviewers' remarks. Please consider and address each of the comments raised by the reviewers before resubmitting the manuscript. This letter should not be construed as implying acceptance, as a revised version will be subject to re-review.

We look forward to receiving your revised manuscript.

Kind regards,

Wisit Cheungpasitporn, MD

Academic Editor

PLOS ONE

Journal Requirements:

2. In your ethics statement in the manuscript and in the online submission form, please provide additional information about the patient records used in your retrospective study.

Specifically, please ensure that you have discussed whether all data were fully anonymized before you accessed them.

Reviewers' comments:

Reviewer's Responses to Questions

**Comments to the Author**

1. Is the manuscript technically sound, and do the data support the conclusions?

Reviewer #1: Yes

Reviewer #2: Yes

Reviewer #3: Partly

2. Has the statistical analysis been performed appropriately and rigorously? 

Reviewer #1: Yes

Reviewer #2: Yes

Reviewer #3: I Don't Know

3. Have the authors made all data underlying the findings in their manuscript fully available?

Reviewer #1: Yes

Reviewer #2: Yes

Reviewer #3: Yes

4. Is the manuscript presented in an intelligible fashion and written in standard English?

Reviewer #1: Yes

Reviewer #2: Yes

Reviewer #3: Yes

5. Review Comments to the Author

Reviewer #1: The Authors are used Standard English that easily understandable for layman, Appropriate statistical tests are used to fine the association and prevalence of disease, Technically showing sound relation between data type and statistical tests.

Reviewer #2: The manuscript is a good scientific sound. In addition, the manuscript can enplane more results and figures to show the behavior of data. Please, see the attached file, I put my comments on it.

best regards

Reviewer #3: Review for PLoS ONE

Sept. 27, 2020

Title: Risk of Gastrointestinal Cancer in Patients with an Elevated Level of Gamma-glutamyltransferase: A Nationwide Population-Based Study

This paper presents a study assess whether an elevated serum gamma-glutamyltransferase (GGT) level is associated with the risk of gastrointestinal cancer through a large-scale study using nationwide data. In addition, we evaluated the effect of glycemic status on the association between an elevated GGT level and the risk of gastrointestinal cancer, and the contribution of the interaction between these two factors on the incidence of gastrointestinal cancer. However, there are questions that limit my enthusiasm of the paper, as outlined below.

1. Data collection and study population:

a. What is the category for drinker? Is it heavy vs not-heavy?

b. Why did authors consider the quartile level to categorize the GGT level? Instead of four categories, while not only consider two categories < Q4 and > Q4 (e.g., Table 4)? Using any threshold to categorize data can have considerable effect on the distribution of data. Please clarify this part.

2. Statistical method:

a. Authors mentioned the t-test was done, but I couldn’t follow across the manuscript for which analysis the t-test was considered. Please clarify that part along with the more details about being two-sided or one-sided? Equal or unequal variance?

b. Why two Cox models were fitted under each analysis? In addition to that did authors compare the two fitted Cox models using analysis of deviance?

c. BMI variable wasn’t introduced in data collection and just used its abbreviation at method section.

3. Results:

a. Table 1: Why not adding similar table as Suppl. To stratify GGT Q1-4 by sex (M/F)? I assume the reported P value is based on ANOVA test (continuous variables) and Chi-squared test (categorical). I could follow for which test t-test was done? (back to comment 2(a))

b. Table 2, what is the reported P for trend?

c. Instead of reporting P < 0.001, why not to report the exact P value using scientific notation?

6. PLOS authors have the option to publish the peer review history of their article (what does this mean?). If published, this will include your full peer review and any attached files.

Reviewer #1: **Yes: **Khalid Mahmood Anjum

Reviewer #2: **Yes: **Faris Mahdi Alwan

Reviewer #3: No

---

## [Author Response · Author response to Decision Letter 0]

3 Nov 2020

We submitted a separate file of the name "response to reviewer" in submitting the revised manuscript. This file contains our comments on each reviewer's comments. Please refer to the file attached.

---

## [Decision Letter · Decision Letter 1]

22 Dec 2020

Risk of Gastrointestinal Cancer in Patients with an Elevated Level of Gamma-glutamyltransferase: A Nationwide Population-Based Study.

PONE-D-20-12359R1

Dear Dr. Lee,

We’re pleased to inform you that your manuscript has been judged scientifically suitable for publication and will be formally accepted for publication once it meets all outstanding technical requirements.

Kind regards,

Wisit Cheungpasitporn, MD

Academic Editor

PLOS ONE

Additional Editor Comments (optional):

Revised Manuscript is well written. All comments have been addressed and thus accepted for publication.

Reviewers' comments:

Reviewer's Responses to Questions

**Comments to the Author**

1. If the authors have adequately addressed your comments raised in a previous round of review and you feel that this manuscript is now acceptable for publication, you may indicate that here to bypass the “Comments to the Author” section, enter your conflict of interest statement in the “Confidential to Editor” section, and submit your "Accept" recommendation.

Reviewer #1: All comments have been addressed

Reviewer #2: (No Response)

Reviewer #3: All comments have been addressed

2. Is the manuscript technically sound, and do the data support the conclusions?

Reviewer #1: Yes

Reviewer #2: Yes

Reviewer #3: Yes

3. Has the statistical analysis been performed appropriately and rigorously? 

Reviewer #1: Yes

Reviewer #2: Yes

Reviewer #3: Yes

4. Have the authors made all data underlying the findings in their manuscript fully available?

Reviewer #1: Yes

Reviewer #2: Yes

Reviewer #3: Yes

5. Is the manuscript presented in an intelligible fashion and written in standard English?

Reviewer #1: Yes

Reviewer #2: Yes

Reviewer #3: Yes

6. Review Comments to the Author

Reviewer #1: The author use standard English for writing the manuscript, appropriate and valid statistical test are used. The manuscript may accept and publish

Reviewer #2: No comment

Reviewer #3: All comments (related to the previous review) have been addressed. Thank you!

I have only two comments regarding the testing part.

Authors assumed the one-sided and also equal variance assumptions, however both of these key assumptions are strong. Authors didn't do any justification for the equality of variances.

In addition to that, I was asked about the comparison between two adjusted models. I totally understand the reason of doing adjustment for variables, but the point of asking that comparison was you can get better idea which model performs better. Adding more variables to do adjustment is not necessarily provide better model or better to say most powerful model.

I refer these two comments to editor and I would like to ask editor to follow up with these two comments.

Thank you

7. PLOS authors have the option to publish the peer review history of their article (what does this mean?). If published, this will include your full peer review and any attached files.

Reviewer #1: **Yes: **Khalid Mahmood Anjum

Reviewer #2: **Yes: **Dr. Faris Mahdi Alwan

Reviewer #3: No

---

## [Editor Report · Acceptance letter]

22 Jan 2021

PONE-D-20-12359R1 

Risk of gastrointestinal cancer in patients with an elevated level of gamma-glutamyltransferase: A nationwide population-based study 

Dear Dr. Lee:

I'm pleased to inform you that your manuscript has been deemed suitable for publication in PLOS ONE. Congratulations! Your manuscript is now with our production department. 

Kind regards, 

on behalf of

Dr. Wisit Cheungpasitporn 

Academic Editor

PLOS ONE